# SCD2 Regulation Targeted by miR-200c-3p on Lipogenesis Alleviates Mesenchymal Stromal Cell Senescence

**DOI:** 10.3390/ijms25158538

**Published:** 2024-08-05

**Authors:** Xiao Yu, Chang Zhang, Qianhui Ma, Xingyu Gao, Hui Sun, Yanan Sun, Yuezeng Wang, Haiying Zhang, Yingai Shi, Xiaoting Meng, Xu He

**Affiliations:** 1The Key Laboratory of Pathobiology, Ministry of Education, College of Basic Medical Sciences, Jilin University, Changchun 130021, China; yuxiao23@mails.jlu.edu.cn (X.Y.); czhang21@mails.jlu.edu.cn (C.Z.); mqh2524115611@163.com (Q.M.); xingyug20@mails.jlu.edu.cn (X.G.); sunhui0729sh@163.com (H.S.); sunyanansunnysmile@163.com (Y.S.); yuezeng@jlu.edu.cn (Y.W.); zhanghaiy@jlu.edu.cn (H.Z.); shiya@jlu.edu.cn (Y.S.); 2Department of Histology & Embryology, College of Basic Medical Sciences, Jilin University, Changchun 130021, China; mengxt@jlu.edu.cn

**Keywords:** senescence, mesenchymal stromal cells, SCD2, lipogenesis, miR-200c-3p

## Abstract

The senescence of bone marrow mesenchymal stromal cells (MSCs) leads to the impairment of stemness and osteogenic differentiation capacity. In a previous study, we screened out stearoyl-CoA desaturase 2 (SCD2), the most evidently changed differential gene in lipid metabolism, using combined transcriptomic and metabolomic analyses, and verified that SCD2 could mitigate MSC senescence. However, the underlying molecular mechanism by which the rate-limiting enzyme of lipogenesis SCD2 manipulates MSC senescence has not been completely understood. In this study, we demonstrate that SCD2 over-expression alleviates MSC replicative senescence and ameliorates their osteogenic differentiation through the regulation of lipogenesis. Furthermore, SCD2 expression is reduced, whereas miR-200c-3p expression is elevated in replicative senescent MSCs. SCD2 is the direct target gene of miR-200c-3p, which can bind to the 3′-UTR of SCD2. MiR-200c-3p replenishment in young MSCs is able to diminish SCD2 expression levels due to epigenetic modulation. In addition, SCD2-rescued MSC senescence and enhanced osteogenic differentiation can be attenuated by miR-200c-3p repletion via suppressing lipogenesis. Taken together, we reveal the potential mechanism of SCD2 influencing MSC senescence from the perspective of lipid metabolism and epigenetics, which provides both an experimental basis for elucidating the mechanism of stem cell senescence and a novel target for delaying stem cell senescence.

## 1. Introduction

Aging is a multifactorial process characterized by decreased physiological activity, low levels of energy metabolism, and loss of homeostasis induced by stress responses, resulting in increased risk of diseases and death [1], and its manifestations vary from cell biological function to gene transcription and post-translational modification of proteins [2]. Stem cell exhaustion is considered to be one of the factors influencing aging and contribute to a progressive decline in tissue maintenance and repair [3]. Despite this knowledge, key regulators of aging and a detailed interpretation of the biological mechanisms of aging have remained elusive.

Mesenchymal stromal cells (MSCs) are a class of adult stem cells with high self-renewal and multi-differentiation potential. They are easy to culture in vitro, able to contact with host cells, and possess an immune tolerance, which makes them ideal seed cells for tissue engineering and the treatment of various diseases [4]. However, in the process of in vitro amplification and culture, MSCs display the characteristic phenotypes of cellular senescence, such as morphological changes, attenuated proliferative activity, and imbalance of physiological function [5]. In addition, during the replicative senescence, elevated levels of reactive oxygen species (ROS) can induce damage to the cellular macromolecules, which are involved in several cellular pathways, including P53, FOXO1, Nrf2, microRNAs, and long non-coding RNAs [6]. In hematopoietic stem cells, satellite cells, and neural stem cells, the expression of P16^INK4A^, as an important marker of cellular senescence, increases with age. P16^INK4A^ inhibition can delay the senescence of stem cells and improve the function of stem cells [7]. Therefore, it is very urgent to explore the mechanism of stem cell senescence for stem cell expansion in vitro, aging reversal, and clinical disease treatment.

Aging is the greatest risk factor for the occurrence and development of a range of diseases, such as cardiovascular diseases, osteoporosis, and metabolism-related diseases. Some studies have reported that metabolism lies at the intersection of multiple age-related pathways such as mTOR, AMPK, and anti-aging interventions [8]. Metabolic disorders are the vital hallmarks of aging, and cells can produce lipids through de novo lipogenesis, a process that produces fatty acids from acetyl CoA, malonyl CoA, and NADPH [9]. Intracellular lipids can stimulate the proliferation of stem cells; for example, the enhanced activity of lysophosphatidylcholine acyltransferase 3 (Lpcat3) can inhibit the excessive proliferation of intestinal stem cells (ISCs). Fatty acid synthase (FASN) and acetyl CoA carboxylase (ACC) are two key enzymes in lipid synthesis, and their compromised activity leads to the declined synthesis of FAs and a reduced proliferative capacity of neural stem and progenitor cells (NSPCs) [10]. Therefore, lipid synthesis can contribute to the activation or differentiation of stem cells. Stearoyl-CoA desaturase (SCD) is another key enzyme in lipogenesis, which acts as a switch between fatty acid storage and consumption, promoting and ameliorating lipid diseases [11]. SCD is closely related to stem cell proliferation and osteogenic differentiation [12]. In a rat model of type 2 diabetes, the underdifferentiation of MSCs into osteoblasts gave rise to inflammation and diminished fracture healing ability, while the over-expression of SCD1 facilitated osteogenic differentiation of MSCs, thus accelerating fracture healing [13]. Nevertheless, how the alterations in lipid metabolism render stem cell senescence remains unexplored to date. A study revealed via a lipidomic analysis that the alterations in lipid profiles may result in the disrupted differentiation capacity of senescent MSCs, manifested by increased adipocytes and decreased osteoblasts. Hence, exploring the effects of abnormal lipid homeostasis on stem cell function in the process of cellular senescence and its underlying mechanisms will favor the identification of new targets for the treatment of age-associated diseases [14]. Our previous studies have shown that over-expression of SCD2 can rejuvenate the senescent MSCs [15], but the mechanism by which SCD2 affects stem cell senescence has not been illuminated.

Epigenetic mechanisms, including DNA methylation, histone modification, and microRNA (miRNA), can produce heritable phenotypic changes without altering the DNA sequence. Non-coding RNAs (ncRNAs) are the crucial regulatory factors for gene and protein expression. MicroRNAs, the most characteristic class of ncRNAs, consist of about 22 nucleotides, which can modulate gene expression at the post-transcriptional level by binding directly to the 3′ untranslated region (3′-UTR) of the target mRNAs [16]. MiRNAs are involved in the manipulation of cellular senescence and the occurrence of age-related diseases such as cancer. The down-regulation of miR-124-3p, miR-129-5p, and miR-378 can facilitate the expression levels of lipid metabolism-related genes in malignant prostate cells [17]. Several miRNAs are known to participate in the modulation of cholesterol homeostasis and lipoprotein metabolism, such as miR-148a, miR-128-1, miR-483, miR-520d, miR-224, miR-30c, miR-122, miR-33, miR-144, and miR-34. Among them, miR-128-1 is able to exert a regulatory effect on the genes involved in lipid synthesis, such as FASN, SIRT1, and lysine deacetylase [18]. Additionally, FAS and SCD act as the vital regulators of fatty acid synthesis in the liver, and miR-212-5p restrains lipid accumulation in hepatocytes by down-regulating their expression [19]. MiR-145 is one of the miRNAs that regulate lipid metabolism in the liver, as well as in other tissues and organs [20]. Statins such as atorvastatin are able to up-regulate miR-145-5p expression in hepatocytes by activating the PI3K/AKT pathway [21]. Some studies have shown that the expression levels of miR-31-5p, miR-141-3p, and miR-200c-3p are elevated with age [22]. Among them, miR-141-3p and miR-200c-3p belong to the same family [23]. The suppression of miR-141-3p expression can alleviate the senescence of MSCs, while the effect of miR-200c-3p on MSC senescence has not been elucidated.

In this study, a bioinformatics analysis reveals that SCD2 is a potential target for miR-200c-3p. Since the inverse expression level between miRNA and mRNA implies a possible inhibitory mechanism, we attempted to unravel whether the regulation of SCD2 on MSC senescence was targeted by miR-200c-3p. Therefore, the present study focused on the expression changes in miR-200c-3p and SCD2 in young and senescent MSCs and further investigated the underlying mechanism of SCD2 in MSC senescence via miR-200c-3p targeting.

## 2. Results

### 2.1. MSCs at Late Passage Present Obvious Senescent Phenotypes

Mounting evidence has shown that MSCs in culture possess limited proliferative capacity before entering a state of replication arrest, and the senescence of replication arrest can seriously impair the function of MSCs in vitro and in vivo after prolonged culture. In this study, early passage MSCs (EPMSCs) and late passage MSCs (LPMSCs) were obtained by continuous in vitro subculture. Morphologically (Figure 1A), EPMSCs exhibited fibroblast-like morphology with elongated and spindle-shaped cell bodies, while LPMSCs displayed senescence-like morphology with flattened, enlarged, and irregular-shaped cell bodies, weak stereopsis, and visible granules in the cytoplasm. By analyzing the length, width, and surface area of the cells, it was found that compared with EPMSCs, the length–width ratio of LPMSCs decreased (Figure 1B), while the surface area increased (Figure 1C). Senescence-associated β-galactosidase (SA-β-gal) staining is considered to be the gold standard for identifying senescent cells in cell cultures [24]. According to SA-β-gal staining, the number of blue-stained cells in LPMSCs was enhanced compared with EPMSCs (Figure 1D). A quantitative analysis using Image J showed that the positive rate of senescent cells in LPMSCs augmented significantly (Figure 1E). Further, the expression level of senescence-related factor P16^INK4A^ (Figure 1F) and P21^WAF1/CIP1^ (Figure 1G) mRNA in LPMSCs by RT-qPCR was remarkably higher than that in EPMSCs. The above results indicate that the replicative senescent MSCs were successfully obtained via in vitro consecutive cultivation.

### 2.2. SCD2 Expression Is Enforced in Late Passage MSCs

Cellular senescence is accompanied by progressive metabolic changes and functional decline [25]. The energy metabolism of cells mainly originates from carbohydrates, lipids, and proteins. Our previous transcriptomic and metabolomic analyses of young MSCs and senescent MSCs showed significant alterations in metabolism-related pathways, including remarkable changes in the expression levels of genes associated with lipid metabolism. Interestingly, we confirmed by RT-qPCR and Western blot that the expression level of SCD2, a gene related to lipogenesis, was most significantly reduced [15]. To corroborate the functional role of SCD2 in MSC senescence, we infected LPMSCs with lentivirus over-expressing SCD2 (LV-SCD2) and lentivirus vector (LV-Vector), respectively. Fluorescence microscopy showed that the cells in both groups expressed green fluorescent protein (Figure 2A). RT-qPCR and Western blot were further applied to detect the infection efficiency of SCD2 lentivirus. The results revealed that the expression level of SCD2 mRNA was up-regulated approximately 3-fold (Figure 2B), and the level of SCD2 protein was elevated about 2-fold in the LV-SCD2 group compared with the Vector group (Figure 2C), suggesting that SCD2 was successfully over-expressed in LPMSCs.

### 2.3. SCD2 Replenishment Alleviates MSC Senescence by Boosting Lipogenesis

To determine whether cellular senescence could be rescued by enforcing SCD2 expression in LPMSCs, SA-β-gal staining was first performed. The results demonstrated that the number of blue-stained cells in the LV-SCD2 group was reduced compared with the LV-Vector group (Figure 3A). A statistical analysis further testified that compared with the LV-Vector group, the positive rate of senescent cells in the LV-SCD2 group was extraordinarily abolished (Figure 3B). Moreover, the expression level of age-related factor P16^INK4A^ mRNA was lower in the LV-SCD2 group than in the LV-Vector group, as indicated by the results of RT-qPCR (Figure 3C). The above data manifest that SCD2 repletion can attenuate MSC senescence.

Increasing evidence has shown that MSCs gradually lose their differentiation potential during extended culture, such as a weakened adipose differentiation and complete loss of osteogenic differentiation potential. Transcriptome analyses consistently showed that long-term cultivation of MSCs in vitro resulted in the down-regulation of genes involved in cell differentiation. For example, runt-related transcription factor 2 (Runx2), is an essential osteoblast transcription mediator whose expression levels decline in the process of MSC aging. It was reported that the osteogenic differentiation of senescent MSCs was diminished, while SCD over-expression improved the osteogenic differentiation of MSCs [12]. We thus assessed the effect of SCD2 replenishment on MSC osteogenic differentiation. It was found through alizarin red s staining that compared with the LV-Vector group, the red bone matrix calcium deposition in the LV-SCD2 group was greatly increased (Figure 3D). The results of Western blot further confirmed that the expression of Runx2, an osteogenic differentiation-related protein, was markedly up-regulated in the LV-SCD2 group (Figure 3E). The above data unravel that SCD2 over-expression in replicative senescent MSCs can ameliorate their osteogenic differentiation. Then, it is speculated that SCD2 may improve MSC osteogenic differentiation by repressing cellular senescence.

Since SCD2 is a rate-limiting enzyme for lipogenesis, enhanced SCD2 expression may influence the intracellular triglyceride content. Consequently, we next detected the intracellular triglyceride content in SCD2-over-expressed MSCs by Nile red staining. As shown in Figure 3F, compared with the LV-Vector group, MSCs in the LV-SCD2 group had stronger red fluorescence. The quantitative analysis results further verified that the intracellular triglyceride content in LV-SCD2 group was significantly elevated. Taken together, SCD2 over-expression can facilitate the lipogenesis of senescent MSCs, indicating that SCD2 repletion may relieve MSC senescence by favoring lipid synthesis.

### 2.4. Prediction and Identification of Upstream miRNAs Targeting SCD2

Messenger RNA (mRNA) is regulated by multiple miRNAs that degrade or inhibit its translation to proteins by interacting with the 3′-UTR [26]. The over-expression of SCD2 can attenuate MSC senescence by facilitating lipid metabolism; we thus surmised that SCD2 mRNA may have potential complementary binding sites with some miRNAs. Consequently, we further selected the top two miRNAs through bioinformatics online prediction websites, miR-199a-3p and miR-200c-3p, respectively (Figure 4A). Studies have shown that the miR-141/200c over-expression in mature T-cell leukemia/lymphoma cell lines can accelerate proliferation and reduce stress-induced cell death [27]. In our study, RT-qPCR verification showed that compared with EPMSCs, the expression levels of both miRNAs were elevated in LPMSCs, especially miR-200c-3p, which was more significantly up-regulated (Figure 4B). As a result, miR-200c-3p was selected for subsequent mechanism exploration. 

Since the expression of miR-200c-3p was remarkably enhanced in LPMSCs, we enforced miR-200c-3p expression in EPMSCs by miR-200c-3p mimic transfection. Then, the efficiency of miR-200c-3p over-expression and SCD2 expression were both detected. RT-qPCR results demonstrated that the expression level of miR-200c-3p was dramatically up-regulated in EPMSCs transfected with the miR-200c-3p mimic (Figure 4C), while the expression levels of SCD2 mRNA (Figure 4D) and protein (Figure 4E) were both obviously down-regulated, that is, miR-200c-3p and SCD2 presented the inverse expression trend. The above results indicate that the over-expression of miR-200c-3p can affect SCD2 expression, suggesting that miR-200c-3p might have a targeted negative regulation on SCD2. Further, a luciferase reporter gene construct was generated, in which SCD2 3′-UTR or a putative miR-200c-3p seed-sequence binding mutation site was inserted behind the luciferase gene. The results of the dual luciferase reporter gene assay showed that compared with the NC group, the over-expression of miR-200c-3p strikingly reduced the luciferase activity of r-SCD2-WT (Figure 4F), while over-expression of miR-200c-3p had no significant impact on the luciferase activity of r-SCD2-MUT. Consequently, miR-200c-3p can bind to the 3′-UTR of SCD2, which is the direct target gene of miR-200c-3p, and negatively modulate SCD2 expression.

### 2.5. MiR-200c-3p Over-Expression Accelerates MSC Senescence and Eliminates Lipogenesis

With respect to the high levels of miR-200c-3p expression exhibited in replicative senescent LPMSCs, we then investigated whether miR-200-c-3p over-expression in EPMSCs could accelerate MSC senescence. As indicated in Figure 5A, compared with the NC group, the number of blue-stained cells in the miR-200c-3p mimic group was increased, and the positive rate of SA-β-gal in cells was significantly elevated. Meanwhile, the expression of P16^INK4A^ mRNA in the miR-200c-3p mimic group was obviously higher than that in the control group (Figure 5B). Western blot results manifested that after miR-200c-3p repletion, the expression of Runx2, a marker protein of osteoblasts, was particularly lower than that of the NC group (Figure 5C), indicating that miR-200c-3p up-regulation could attenuate the osteogenic differentiation of MSCs. Nile red staining further displayed that the intracellular red fluorescence was weakened in the miR-200c-3p mimic group as compared to the NC group (Figure 5D). Quantitative analysis revealed that the intracellular triglyceride content in the miR-200c-3p mimic group was extraordinarily diminished. These results suggest that the over-expression of miR-200c-3p can aggravate MSC senescence and abolish the lipid synthesis of MSCs.

### 2.6. SCD2 Is Targeted by miR-200c-3p to Influence MSC Senescence via Regulating Lipogenesis

In order to explore whether SCD2 was targeted by miR-200c-3p to regulate MSC senescence by affecting lipid synthesis, we conducted a miR-200c-3p/SCD2 functional rescue assay in LPMSCs. As shown in Figure 6, SCD2 sufficiency significantly increased the expression levels of SCD2 mRNA and protein, while the co-expression of SCD2 and miR-200c-3p (LV-SCD2+mimic) could remarkably down-regulate the elevation in SCD2 mRNA (Figure 6A) and protein expression levels (Figure 6B) caused by SCD2 sufficiency.

As indicated in Figure 7A, the number of blue-stained cells in LPMSCs was diminished in the LV-SCD2 group compared with the LV-Vector group, while the number of blue-stained cells was obviously enhanced after co-transfection of the miR-200c-3p mimic with LV-SCD2, which was able to reverse the reduction in the positive ratio of senescent cells due to SCD2 repletion. Moreover, the expression of senescence-associated factor P16^INK4A^ mRNA was detected by RT-qPCR in four groups of cells. It was found that the expression level of P16^INK4A^ mRNA was down-regulated after SCD2 over-expression, while the co-expression of miR-200c-3p with SCD2 rescued the decreased level of P16^INK4A^ mRNA (Figure 7B). The above data suggest that miR-200c-3p has a definite targeted suppression on SCD2, thus influencing MSC senescence. The results of alizarin red s staining showed that red calcium salt deposition was augmented in the LV-SCD2 group compared with the LV-Vector group, while red calcium salt deposition notably declined in the LV-SCD2+mimic group (Figure 7C). The expression level of Runx2 was further examined by Western blot, and the results displayed that Runx2 expression level was facilitated in the LV-SCD2 group compared with the LV-Vector group, while miR-200c-3p co-expression with SCD2 was able to down-regulate the elevated expression of Runx2 protein resulted from SCD2 replenishment (Figure 7D). Furthermore, Nile red staining results demonstrated that intracellular red fluorescence was significantly enhanced and intracellular triglyceride content was remarkably boosted in the LV-SCD2 group compared with the LV-Vector group, while miR-200c-3p co-expression with SCD2 led to a notable reduction in intracellular red fluorescence intensity and triglyceride content caused by SCD2 over-expression (Figure 7E). Thus, miR-200c-3p can target and negatively regulate SCD2 to restrain lipid synthesis in MSCs. Taken together, the effect of SCD2-rescued MSC senescence and enhanced lipogenesis can be attenuated by miR-200c-3p replenishment.

## 3. Discussion

Along with individuals’ age, the number of MSCs and their biological activity decrease [28]. Large amounts of neutral amino acids such as valine, isoleucine, and glycine may be used as alternative energy sources to maintain energy homeostasis in senescent cells [29]. In contrast, stem cell proliferation is highly dependent on oxidative phosphorylation and susceptible to oxidative damage and cellular dysfunction. Therefore, scavenging ROS or over-expressing Nrf2, a transcription factor of antioxidant stress response, can contribute to the diminishment in the hyperproliferative phenotypes of senescent intestinal stem cells [30]. The loss of mitochondria number in stem cells may also give rise to senescence-related changes in stem cell function. The over-expression of the mitochondrial regulator PGC-1 in Drosophila effectively inhibited stem cell senescence and extended the average lifespan of Drosophila [31]. In addition, the tricarboxylic acid cycle (TCA) in mitochondria is a key hub for integrating metabolism and various signals in the senescence network, and metabolites such as nicotinamide adenine dinucleotide, α-ketoglutarate, and β-hydroxybutyrate in the TCA cycle play central roles in senescence-related signaling and metabolic dysregulation [32]. The increased levels of choline, phosphatidylcholine, and phosphatidylethanolamine in senescent MSCs may be caused by cell membrane damage due to elevated ROS. Most lipids, such as polyunsaturated fatty acids, are susceptible to oxidation caused by oxygen free radicals, which in turn leads to cellular senescence [33]. Tumor cells can also grow rapidly and survive through lipid metabolic reprogramming [34]. It has been reported in the literature that the anti-aging mechanisms in *Cryptobacterium hidradenum* and rats may be related to their ability to manipulate lipid metabolism [35]. Lipids modulate the biological properties of stem cells by affecting energy storage, plasma membrane composition, signal transduction, and gene expression. In conclusion, in the senescence process of MSCs, the antioxidant capacity of cells is reduced, and metabolites related to lipid metabolism are significantly altered.

Monounsaturated fatty acids are essential for the maintenance of normal epidermal permeability barrier function and lipid biosynthesis. The formation of some biological membrane systems such as cell membranes, mitochondrial membranes, and endoplasmic reticulum membranes is closely correlated with lipid metabolism [36]. SCD2 is the rate-limiting enzyme that catalyzes the synthesis of monounsaturated fatty acids and is involved in lipid synthesis during biological development [37]. It has been reported that SCD is associated with the development of age-related diseases. For instance, the incidence of age-related macular degeneration has an intimate relationship with the abnormal expression of SCD2, which manipulates macrophage-mediated inflammatory responses and pathological angiogenesis [38]. A high expression level of SCD1 promotes the proliferation, migration, and invasion of a variety of cancer cells and suppresses cell apoptosis [39]. Additionally, in patients with diabetes and femoral head necrosis, SCD over-expression facilitates the osteogenic differentiation of MSCs [13].

MSCs predispose to senesce with advancing age, and their osteogenic differentiation capacity is compromised. Currently, it has been reported that SCD is associated with MSC osteogenic differentiation, but the definite relationship between them has not been explored in depth. In this study, we first screened the senescence-related lipid metabolism gene SCD2 in MSCs based on a preliminary multi-omics analysis. Then, MSCs over-expressing SCD2 by lentivirus infection were successfully constructed by the observation of a green fluorescent protein under a fluorescence microscope and RT-qPCR and Western blot verification. We found that SCD2 over-expression not only inhibited MSC senescence but also promoted MSC osteogenic differentiation. Meanwhile, SCD2 repletion increased the triglyceride content and enhanced lipid synthesis in senescent MSCs. This is consistent with literature reports that SCD can contribute to lipid synthesis [40] and osteogenic differentiation [41]. On this basis, we propose for the first time that SCD2 has an inhibitory effect on MSC senescence. In conclusion, we suggest that SCD2 can delay MSC senescence and facilitate their osteogenic differentiation by boosting cellular lipid metabolism.

In the present study, we found that SCD2, a gene related to lipid metabolism, could modulate cellular lipid synthesis and thus influence MSC senescence, but its specific mechanism needs to be further investigated. Numerous studies have shown that miRNAs are closely related to a variety of physiological and pathological processes in the organism, including cellular homeostasis, proliferation, differentiation, migration, apoptosis, and multiple diseases [42]. One study manifested that miR-141-3p expression in human and mouse bone marrow stromal cells was heightened with age. There were 22 age-related differentially expressed miRNAs in the aging rat pituitary gland, of which miR-141-3p affected pituitary aging by declining pituitary growth hormone, and miR-141-3p was involved in regulating the senescence process in a variety of cells [43]. In contrast to the declined expression of Nampt, miR-34a expression was increased in senescent MSCs. MiR-34a over-expressed young MSCs presented age-related features, such as senescent morphology, prolonged cell proliferation, decreased osteogenic differentiation efficacy, increased age-related β-galactosidase activity, and elevated expression of age-related factors [26]. MiRNAs are considered to be key regulators in delaying cellular senescence and extending lifespan [44]. We thus hypothesized that certain miRNAs instigated stem cell senescence by targeted-regulating SCD2 in the senescence process of MSCs. Through miRBase, Targetscan, and other databases, we screened the two top scoring miRNAs, miR-200c-3p and miR-199a-3p, which might participate in the regulation of stem cell senescence by directly targeting SCD2. Validated by RT-qPCR, the expression of miR-200c-3p was found to be more significantly up-regulated in senescent MSCs, so miR-200c-3p was used as a target to explore the mechanism of SCD2 regulating MSC senescence.

MiR-141-3p and miR-34a were reported to exert a regulatory role in MSC senescence by influencing glucose metabolism [45], while miR-141-3p, miR-200a-3p, miR-200b-3p, and miR-200c-3p are four members of the miR-200 family [46]. The relationship between all the above four miRNAs and aging has been reported, but the role of miR-200c-3p on aging has been inconsistently stated. One study discovered that miR-200c-3p promoted the proliferation of adipose-derived MSCs and delayed cellular senescence [47]. The other study demonstrated that the expression levels of miR-31-5p, miR-141-3p, and miR-200c-3p were up-regulated with age in the liver [22]. MiR-200c directly targeted ZEB1 to induce apoptosis in MSCs [48]. Increased miR-200c expression under oxidative stress conditions modulated the SIRT/FOXO1/eNOS pathway by inducing ROS production [49]. In the present study, we found that miR-200c-3p expression was up-regulated in senescent MSCs, whereas SCD2 expression was decreased, suggesting that there might be an adverse correlation between miR-200c-3p and SCD2 expression. Further results testified that miR-200c over-expression notably repressed SCD2 expression in young MSCs and dual luciferase reporter gene assays confirmed that miR-200c-3p was able to bind to the 3′-UTR of SCD2 mRNA, implying that SCD2 is a direct target gene for miR-200c-3p.

We further investigated the effect of miR-200c-3p on cellular senescence and lipid synthesis in MSCs. MiR-200c-3p repletion not only suppressed cellular lipid synthesis and promoted MSC senescence but also diminished the MSC osteogenic differentiation ability. These results above are consistent with miR-200c over-expression inducing growth arrest, apoptosis, and senescence in human umbilical vein endothelial cells [23], suggesting that miR-200c-3p possesses a role in regulating cellular lipid synthesis and affecting MSC senescence. To clarify the deeper underlying mechanism by which miR-200c-3p/SCD2 manipulates MSC senescence, we performed a functional rescue assay. The results confirmed that miR-200c-3p co-expression with SCD2 could alleviate the up-regulation of SCD2 levels, the lipogenesis and osteogenic differentiation induced by SCD2 replenishment, and reverse the inhibitory effect of SCD2 over-expression on MSC senescence. This indicates that SCD2 can be directly targeted by miR-200c-3p to modulate cellular lipid synthesis, which in turn affects the senescence of MSCs. However, there are also some limitations in the present study, such as the lack of miR-200c-3p-inhibitor rescue experiments to strengthen the conclusion, which deserves deeper exploration in the future.

Taken together, our data validate the roles of the differential lipid metabolism gene SCD2 associated with MSC senescence. SCD2 over-expression can promote lipid synthesis and thus rescue MSC senescence, which is manipulated by miR-200c-3p targeting (Figure 8). This study reveals the molecular mechanism of miR-200c-3p/SCD2 regulation MSC senescence from the perspective of lipid metabolism and epigenetics, which provides a novel target and an important experimental basis for delaying the senescence of stem cells and preventing age-related diseases.

## 4. Materials and Methods

### 4.1. MSC Isolation and Culture

Healthy male Wistar rats with SPF grade of 150 g to 180 g were placed in a quiet and stable state, euthanized by CO_2_ inhalation, and sterilized. After that, the femur, tibia, and humerus of the rats were quickly separated, which were further placed in a 50 mL centrifuge tube and soaked with PBS containing a 1% penicillin/streptomycin solution and then quickly moved to the ultra-clean hood. After the muscle and tissue on the bone surface were removed, they were soaked in a complete culture medium. Subsequently, the bone marrow cavity at both ends of the bone was opened with bone-chewing forceps, and the culture medium was extracted with a 5–10 mL syringe to flush the bone marrow cavity. After rinsing 3–4 times, the bone marrow cavity became translucent. The rinsed liquid was placed on a cell filter and centrifuged at room temperature at 1500 rpm for 5 min. After that, the cell suspension was re-suspended with a 5 mL pipette and placed in a 6 cm cell culture dish for culture. On the second day, half of the fluid was changed, and the culture was continued. When the cells were fused to about 80% confluence, they were cultivated for about 10 generations through in vitro serial passages. Then, the cellular senescence in different passages was identified, and the replicative senescence model of MSCs was then established. P2-P3MSCs were used as early-passage MSCs (EPMSCs) and P9-P10MSCs were used as late-passage MSCs (LPMSCs).

### 4.2. Morphological Observation and Quantitative Analysis of MSCs

The morphology and growth conditions of EPMSCs and LPMSCs were observed under a high-magnification (×200) microscope, and multiple fields of view were selected separately to capture plots. Cell Entry software (Ver.2.1) was utilized to calculate the cell surface area as well as the ratio of the longest to the shortest axis as the cell aspect ratio. Finally, the obtained values of cell surface area and cell aspect ratio were statistically analyzed.

### 4.3. Senescence-Associated β-Galactosidase (SA-β-Gal) Staining

EPMSCs and LPMSCs were cultured in 6-well plates, respectively. When the cells reached 70% confluence, the cells were washed with PBS 3 times. The activity of SA-β-gal was evaluated with the aging cell histochemical staining kit (Beyotime, Shanghai, China). First, a 500 μL fixing solution was added to each well to fix the cells for 15–18 min, and the cells were then washed with PBS again. After washing, 1 mL of working liquid (including dyeing solution A and B, 10 μL each; dyeing solution C, 930 μL; and X-Gal solution, 50 μL) was added to each well. Next, the plates were sealed with sealing film, plastic wrap, and tin foil to avoid light, and cultivated overnight in a CO_2_-free incubator at 37 °C. Finally, the staining results of positive cells were observed under a phase contrast microscope and photographed, and the rate of SA-β-gal positive cells was statistically analyzed.

### 4.4. Construction of SCD2-Over-Expressed MSCs

LPMSCs were inoculated into the 6-well plate, and different gradients for time and concentration were set according to the cell density to determine the optimal infection conditions. SCD2 over-expressing lentivirus (GeneChem, Shanghai, China), HitransG A/P and complete culture medium without the penicillin/streptomycin solution were added successively, and the plate was cultured in a cell incubator at 37 °C with 5% CO_2_. About 12 h later, the liquid was changed completely, and the culture was continued for 2–3 days. Finally, the expression of the green fluorescent protein was observed under a fluorescence microscope. When the expression of the green fluorescent protein was low, purinamycin could be applied for drug screening to obtain MSCs over-expressing SCD2 with high efficiency.

### 4.5. Osteogenic Differentiation and Alizarin Red s (ARS) Staining

MSCs have a multidirectional differentiation potential, such as osteogenesis. In the present study, alizarin red staining was performed to explore the effect of SCD2 on osteogenic differentiation. Firstly, 1 mL of 0.1% gelatin was added to a 6-well plate, and the plate was gently shaken and incubated in a CO_2_ incubator for at least 30 min. After that, the gelatin was aspirated, and MSCs were then inoculated in 6-well plates, and the cell density was determined according to the growth state. Subsequently, 2 mL of the prepared complete culture medium was added to each well, and the plates were cultivated in a cell culture incubator. Secondly, when the cell confluence reached about 75%, the culture medium was discarded. The osteogenesis-induced differentiation medium was then added to the wells and changed every 2–3 days. After 14–17 days of induction, cells were washed with PBS 2–3 times after discarding the induction solution and fixed in 4% paraformaldehyde solution for 25–30 min at room temperature. Then, the fixative solution was discarded, and the cells were washed with PBS 2–3 times. Finally, a 2 mL alizarin red s working solution was added to each well for 8–10 min. After washing with PBS 2–3 times, the staining results were observed under the microscope to determine the osteogenic differentiation of MSCs.

### 4.6. Nile Red Staining

The storage solution at the concentration of 1 mM was prepared by dissolving Nile red with DMSO and stored away from light. Then, the cells with 70% confluence was fixed in a 4% paraformaldehyde solution at room temperature for 30 min. After that, a 1× Nile red working solution (1 mM storage solution: PBS = 1:1000) was added to each well, and the plate was incubated at 37 °C for 5–10 min away from light. After washing with PBS 2–3 times, a DAPI staining solution was added, and the plate was then incubated at 37 °C away from light for 5–10 min. Eventually, the red fluorescence was observed under the fluorescent microscope with the same exposure time, and the photos were then taken for analyses.

### 4.7. Prediction for Upstream miRNAs of SCD2

The upstream miRNAs of SCD2 were predicted by online bioinformatics sites, including Targetscan (David Bartel Lab, Whitehead Institute for Biomedical Research, Cambridge, MA, USA), miRanda (Memorial Sloan Kettering Cancer Center, New York, NY, USA), PicTar (Rajewsky lab, New York, NY, USA and Max Delbruck Centrum, Berlin, Germany), miRbase (University of Manchester, Manchester, UK), and an online database for miRNA target prediction (microRNA.org). Then, the miRNAs with possible binding sites were screened and further experimentally validated.

### 4.8. Dual Luciferase Reporter Assay

After seeding 293T cells into a 96-well plate at a density of 1 × 10^4^ per well, the culture medium was changed to a DMEM/F12 complete medium without penicillin/streptomycin. A MiR-200c-3p mimic or NC mimic was diluted with 12.5 µL of 1× ribo*FECT*^TM^ CP Buffer, and 1 µL of SCD2 3′ UTR dual luciferase reporter plasmid (WT or Mut) (Ribobio, Guangzhou, China) was added. Then, the mixture was mixed gently. After incubation at 25 °C for 5 min, 1.25 μL of ribo*FECT*^TM^ CP Reagent was added. Next, the mixture was incubated at 25 °C for 15 min and added to each well, and then the plates were incubated. After incubation for 48 h, the plates were balanced at 25 °C for 10 min, and 100 μL of firefly detection reagent was added to each well, followed by thoroughly mixing away from light. The 100 μL Renilla detection reagent (Beyotime, Shanghai, China) was then added to each well. Eventually, the plates were incubated at 25 °C for 10 min. The 96-well plate Luminometer (Promega Glomax, Madison, WI, USA) was used to measure the amount of light released. The relative fluorescence activity of each well was evaluated in terms of RLU using firefly/Renilla.

### 4.9. Construction of miR-200c-3p-Over-Expressed MSCs

EPMSCs were firstly passed into 6-well plates. On the next day, the culture medium was discarded, and miR-200c-3p mimics at different concentrations were added to the cells. Then, the medium was changed after 12–24 h. Cell growth was observed at 24 h, 36 h, 48 h, and 72 h, respectively, to determine the optimal infection conditions for the miR-200c-3p mimic. Further, according to the optimal infection conditions, miR-200c-3p mimics, Ribo FECTTMCP Reagent (Ribobio, Guangzhou, China), and the complete medium without the penicillin/streptomycin solution were added proportionally, and the cells were incubated at 37 °C with 5% CO_2_ for 24–48 h. After that, the cells were collected for subsequent experiments.

### 4.10. Gene Expression Analysis

Total RNA was extracted from MSCs using the QIAzol Lysis Reagent from the miRNeasy Mini kit. For the detection of gene expression, 500 ng of total RNA was used to synthesize cDNA with TransScript All-in-One First-Strand cDNA Synthesis SuperMix for qPCR (Transgen biotech, Beijing, China), and then the relative quantity of genes was measured by 2× Taq SYBR^®^Green qPCR Premix (Innovagene, Changsha, China) in the ABI 7300 Real-Time PCR System (Applied Biosystems, New York, NY, USA). After complementary DNA (cDNA) was synthesized with 1000 ng of miRNA using the All-in-One™ miRNA First-Strand cDNA Synthesis Kit (GeneCopoeia, Rockville, MD, USA), the expression levels of miR-199a-3p and miR-200c-3p were measured by real-time quantitative polymerase chain reaction (RT-qPCR) using miRNA-specific qPCR primers and the All-in-One miRNA RT-qPCR Detection Kit (GeneCopoeia, USA) in a 7300 Real-Time PCR System. Rat-specific primers were synthesized and are listed as follows: β-actin: forward 5′-GGAGATTACTGCCCTGGCTCCTA-3′, reverse 5′-GACTCATCGTACTCCTGCTTGCTG-3′; SCD2: forward 5′-GCAGATGTTCGCCCTGAAATTA-3′, reverse 5′-CAAATATGCAAAGAGGCAGGTGTAG-3′; P16^INK4a^: forward 5′-AACACTTTCGGTCGTACCC-3′, reverse 5′-GTCCTCGCAGTTCGAATC-3′; P21^WAF1/CIP1^: forward 5′-GACATCACCAGGATCGGACAT-3′, reverse 5′-GCAACGCTACTACGCAAGTAG-3′; RsnRNA U6 RmiRQP9003GeneCopoeia, China; rno-miR-199a-3p RmiR6090GeneCopoeia, China; rno-miR-200c-3p RmiR6096GeneCopoeia, China. β-actin and U6 were amplified as the reference genes to normalize the relative expression of mRNA and miRNA using the 2^−ΔΔCt^ cycle threshold method.

### 4.11. Western Blot Analysis

The protein was extracted with a RIPA lysis buffer, and the total protein content was determined by the BCA Protein Assay Kit (Beyotime, Shanghai, China). In total, 30 μg of protein samples was obtained, and the SDS protein sampling buffer and PBS were then added. After the protein samples were boiled for 5–7 min, an 8–12% separating gel and a 5% concentrating gel were prepared according to the molecular weight of the proteins, and electrophoresis was carried out at different voltages. When electrophoresis was completed, the gels were transferred to a PVDF membrane for 40–50 min. Then, the blotted membranes were blocked with 5% non-fat milk for 1–2 h at room temperature and further probed with anti-SCD2 (1:500 dilution, Santa Cruze, Dallas, TX, USA), anti-Runx2 (1:500 dilution Proteintech, Philadelphia, PA, USA), and anti-β-actin (1:2000 dilution, Abcam, Cambridge, UK) diluted in tris-buffered saline (TBS) overnight at 4 °C. After incubating with horseradish peroxidase-conjugated with anti-rabbit IgG secondary antibody (1:2000 dilution, Proteintech, Philadelphia, PA, USA), protein blots were visualized using an enhanced electrochemiluminescence detection system (Amersham Biosciences, Piscataway, NJ, USA). The result was observed after the color development, and the images were saved in TIFF format for further analyses.

### 4.12. Statistical Analysis

All data in this experiment were expressed as mean ± standard deviation, statistical analyses were performed using Graphpad Prism software (VER.8.0.2), and *p* values were obtained by applying an independent-sample *t* test analysis, with statistically significant *p* values indicated thusly: * *p* < 0.05, ** *p* < 0.01, and *** *p* < 0.001.

## Figures and Tables

**Figure 1 ijms-25-08538-f001:**
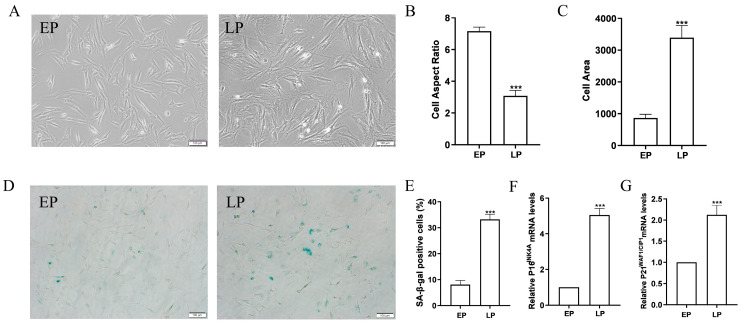
Morphological evaluation of MSCs at different passages. (**A**) Morphological features (scale bar = 100 μm) and quantitative analysis of cell aspect ratio (**B**) and cell area (**C**) of EP (EPMSCs) and LP (LPMSCs). (**D**,**E**) SA-β-gal staining (scale bar = 100 μm) (**D**) and quantitative analysis of the percentages of β-gal-positive cells (**E**) in EP (EPMSCs) and LP (LPMSCs). mRNA expression of the senescence-related factor (**F**) Pl6^INK4A^ and (**G**) P21^WAF1/CIP1^. Data indicate the mean ± SD, *n* = 3. *** *p* < 0.001 vs. EP (EPMSCs).

**Figure 2 ijms-25-08538-f002:**
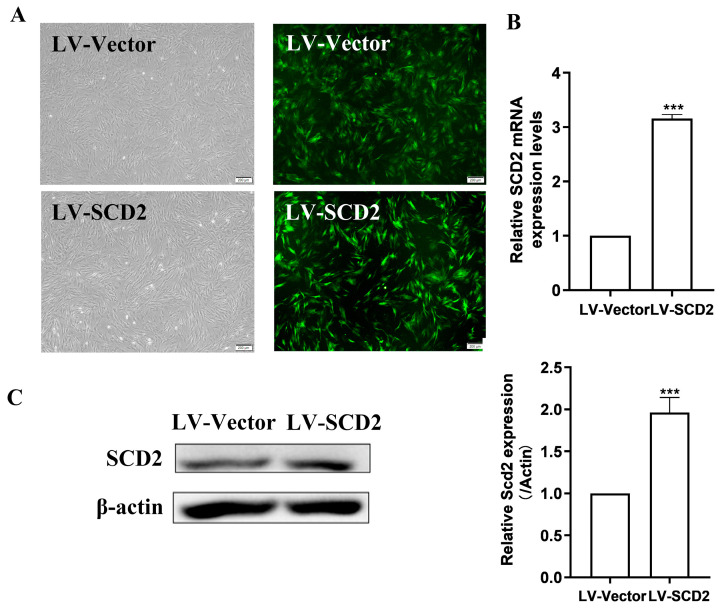
MSCs over-expressing SCD2 were successfully constructed. (**A**) Morphological features and EGFP expression under fluorescence microscope (scale bar = 200 μm). (**B**,**C**) Demonstration of the transduction efficiency of SCD2 over-expression in LP (LPMSCs) by RT-qPCR (**B**) and Western blot (**C**). Data indicate the mean ± SD, *n* = 3. *** *p* < 0.001 vs. LV-Vector.

**Figure 3 ijms-25-08538-f003:**
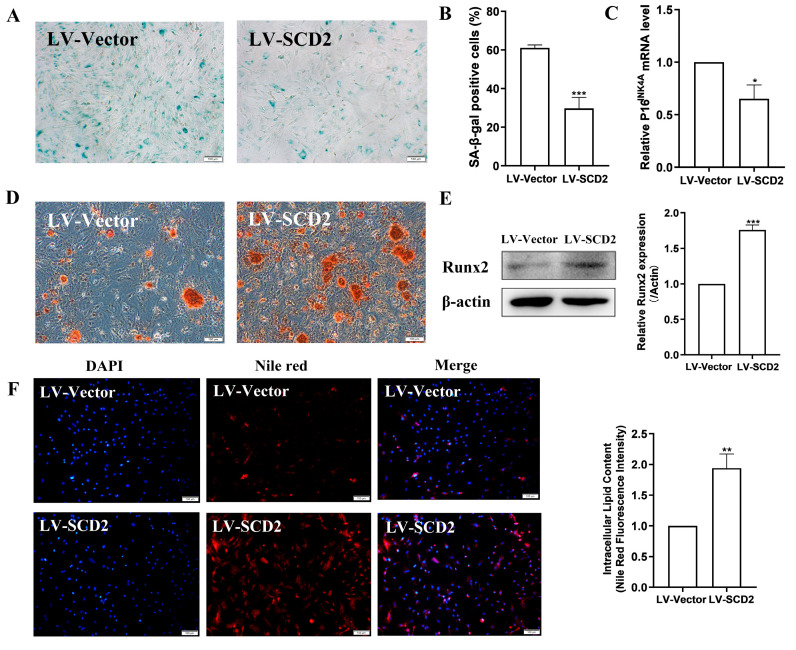
SCD2 over-expression mitigates MSC senescence and promotes lipid synthesis. (**A**,**B**) SA-β-gal staining (scale bar = 100 μm) and quantification (**B**). (**C**) Determination of the expression levels of the senescence marker P16^INK4A^ mRNA after SCD2 repletion by RT-qPCR. (**D**) Alizarin red s staining after osteogenic differentiation of MSCs (scale bar = 100 μm). (**E**) Determination of Runx2 protein expression levels by Western blot. (**F**) Nile red dye (scale bar = 100 μm) and quantification. Data indicate the mean ± SD, *n* = 3. * *p* < 0.05, ** *p* < 0.01, *** *p* < 0.001 vs. LV-Vector.

**Figure 4 ijms-25-08538-f004:**
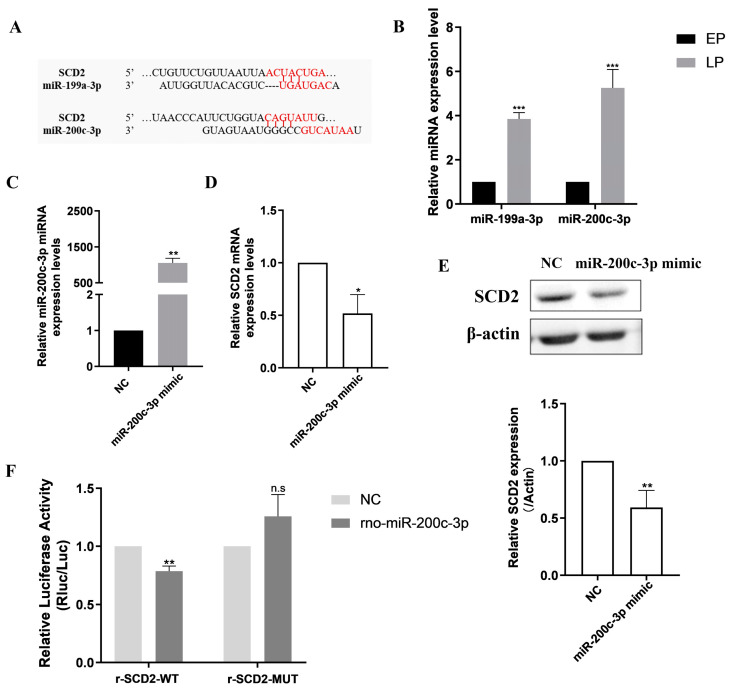
MiR-200c-3p negatively regulates SCD2 expression via directly targeting SCD2. (**A**) Schematic representation of the predicted complimentary base pairing between miR-199a-3p and miR-200c-3p and the 3′-UTR of SCD2 mRNA and the mutated binding site of putative miR-199a-3p and miR-200c-3p seed sequence. (**B**) Detection of SCD2 mRNA expression levels by RT-qPCR after interfering with miR-199a-3p and miR-200c-3p expression. (**C**,**D**) Detection of miR-200c-3p and SCD2 mRNA expression levels by RT-qPCR. (**E**) Detection of SCD2 protein expression levels by Western blot. (**F**) Analysis of luciferase activity in cells co-transfected with miR-NC or miR-200c-3p and plasmid containing the 3′-UTR of the wild type (WT) SCD2 or a mutated (MUT) SCD2 sequence in dual luciferase reporter assay. Data indicate the mean ± SD, *n* = 3. *** *p* < 0.001 vs. EP (EPMSCs), * *p* < 0.05, ** *p* < 0.01, vs. NC (normal control mimic), n.s, not significant.

**Figure 5 ijms-25-08538-f005:**
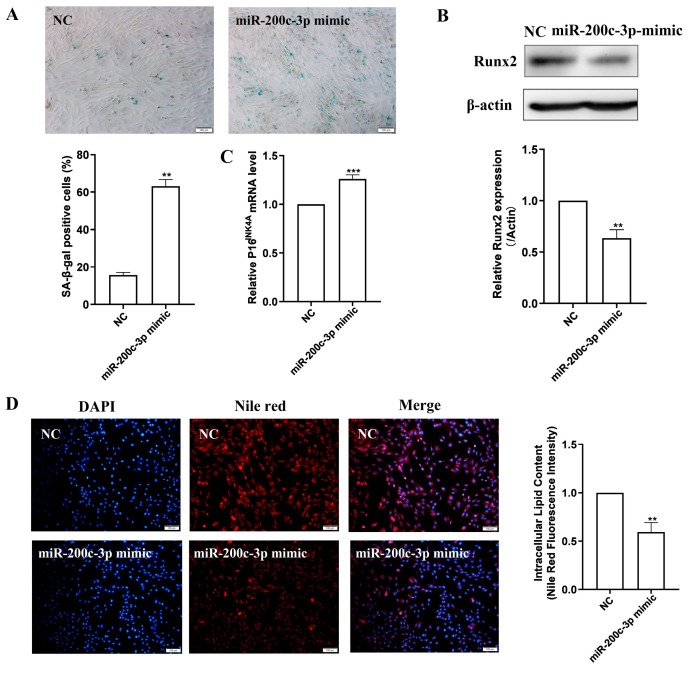
MiR-200c-3p over-expression exacerbates senescence-associated variations and represses lipid synthesis in EP (EPMSCs). (**A**) SA-β-gal staining (scale bar = 100 μm) and quantification. (**B**) Detection of Runx 2 protein expression levels by Western blot. (**C**) Detection of P16^INK4A^ mRNA expression levels by RT-qPCR. (**D**) Nile red dye (scale bar = 100 μm) and quantification. Data indicate the mean ± SD, *n* = 3. ** *p* < 0.01, *** *p* < 0.001 vs. NC (normal control mimic).

**Figure 6 ijms-25-08538-f006:**
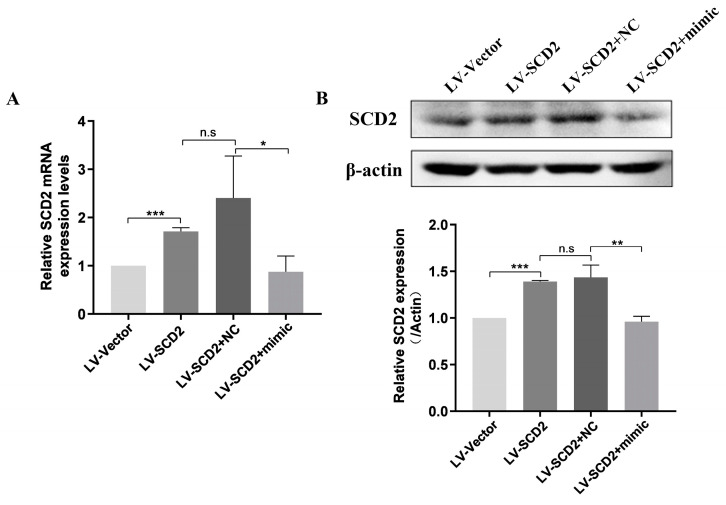
The alterations of SCD2 expression in MSCs after co-expression of miR-200c-3p and SCD2. (**A**) Detection of SCD2 mRNA expression levels by RT-qPCR in LP (LPMSCs) treated with LV-Vector, LV-SCD2, LV-SCD2+NC, and LV-SCD2+mimic. (**B**) Detection of SCD2 protein expression levels by Western blot. Data indicate the mean ± SD, *n* = 3. * *p* < 0.05, ** *p* < 0.01, *** *p* < 0.001 vs. LV-Vector, LV-SCD2, LV-SCD2+NC, n.s, not significant.

**Figure 7 ijms-25-08538-f007:**
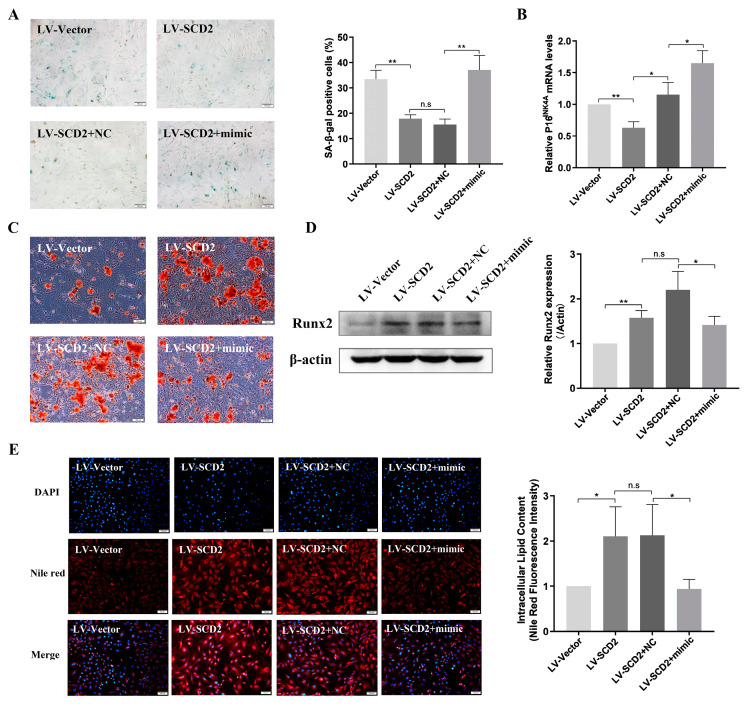
SCD2 targeted by miR-200c-3p influences MSC senescence via modulating lipid synthesis. (**A**) SA-β-gal staining (scale bar = 100 μm) and quantification in LPMSCs treated with LV-Vector, LV-SCD2, LV-SCD2+NC, and LV-SCD2+mimic. (**B**) Detection of mRNA expression levels of senescence-related factors P16^INK4A^ by RT-qPCR after interfering with LV-Vector, LV-SCD2, LV-SCD2+NC, and LV-SCD2+mimic. (**C**) Alizarin red s staining after osteogenic differentiation of MSCs (scale bar = 100 μm). (**D**) Detection of Runx 2 protein expression levels by Western blot. (**E**) Nile red dye (scale bar = 100 μm) and quantification. Data indicate the mean ± SD, *n* = 3. * *p* < 0.05, ** *p* < 0.01 vs. LV-Vector, LV-SCD2, LV-SCD2+NC, n.s, not significant.

**Figure 8 ijms-25-08538-f008:**
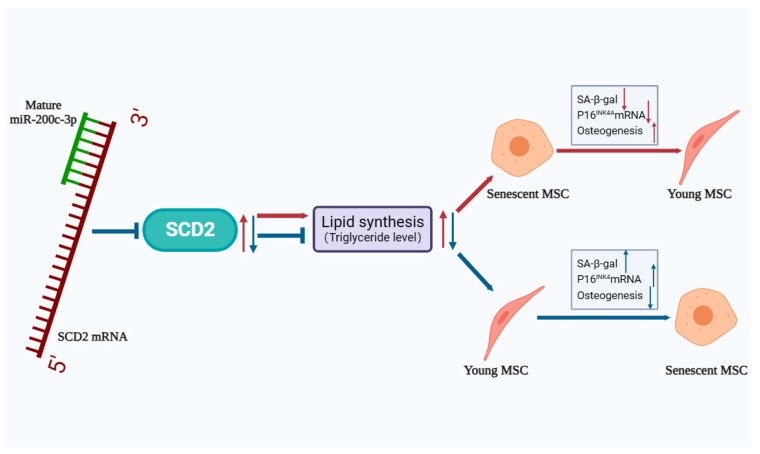
Schematic diagram of miR-200c-3p/SCD2 affecting senescence in MSCs through lipogenesis regulation. SCD2 repletion can facilitate lipid synthesis and thus rescue MSC senescence, and vice versa. The effects of SCD2 mentioned above are negatively manipulated by miR-200c-3p targeting.

## Data Availability

Data contained within the article.

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
