# Peer review of "SCD2 Regulation Targeted by miR-200c-3p on Lipogenesis Alleviates Mesenchymal Stromal Cell Senescence"

_ijms, 2024, doi:10.3390/ijms25158538_

Round 1

Reviewer 1 Report

Comments and Suggestions for Authors

General remarks

ą1/  The study design abstract contains notions about the epigenetic modulations although this is not addressed in the main text.

22/ Results are represented in the Figures using abbreviations without explanation e.g., in Figure 1 and 4: EP, LP (it is understood that this stands for EPMSCs and LPMSCs, respectively); in Figure 4: NC (Normal control? Non-coding?). The authors should simplify readability of the texts using clarification in the Figures or somewhere in the text List of abbreviations (at least main) would be helpful.

Methodology:

  3/ Please provide (translated to English) the animal study protocol and the approval document by the Ethics Committee of Jilin University (protocol code SYXK 2018-0001).

  4/ The design of the study chosen is of questionable value as the relevance of all data are questionable as not always positive and/or negative controls were used. The authors should justify the absence of such control arms providing clear justification that all results are relevant and externally valid.

    5.       The MSC isolation, culture description and osteogenic differentiation descriptions do not allo justify that the cells are really exhibiting multilineage differentiation potential in adipogenic, chondrogenic, and osteogenic cultures.

    6.       Then cell surface area and aspect ratio of the cells obtained in the images were calculated by Cell Entry software for statistical analysis. The precise details of the software and description of aspects are not provided. This should be explained in the text and amended in the text.

     7.       The precise details of SA-beta-Gal staining assay methodology is not provided; thus, this information should be amended in the text. All homemade solutions should be validated.

     8.       General remark - statistical analysis was based on the means although relevance of this methodology without normality test is not acceptable. The Authors should justify the applicability of this test to the rather small sample sizes (N = 3) throughout the work.

      9.       I cannot comment on validity of methods chosen for Prediction for upstream miRNAs of SCD2 (4.7 section), Dual luciferase reporter assay (4.8 section), Construction of miR-200c-3p over-expressed MSCs (4.9 section), and Gene expression analysis (4.10 section) as not sufficient detailed structured information is provided enabling independent replication of the experiments and no justification about the validity of the method (In-house methods? Commercial manufacturers’ method and following instructions?).

Results:

 10.   As this is addressed already in the methodology section, the results obtained are is of questionable value as no positive or negative controls were used. The authors should justify for the each finding for the relevance the results based on external findings by others, if feasible.

 11.   Figures 1, 2, 3, 4, 5, 6, and 7 presents various graphical results (e.g., figure 1 presents result of the cell surface area and aspect ratio of the cells). The sample sizes of 3 is declared whereas the normality of the distribution within this sample size is not obvious. The authors should present individual data in the graphs as separate dots.

 12.   Figure 1 (D) presents visual results of SA-β-gal staining that do not represent values shown in 1 (E) graphs. This should be clarified and corrected.

 13.   Figure 1 (E) shows very high rates of senescence in both EP and LP cells (i/e., ~30% s of SA-β-gal staining in EP and 100% in LP). That observation questions stemness of the cells cultivated. This should be clarified and validity of MSC isolation and identification method corrected.

 14.   Figures 1 (E) do not represent values in 1 (E) graphs. This should be clarified and validity of MSC isolation and identification method corrected.

 15.   Results of SCD2 over-expression in Figure 3B show ~75% of SA-β-gal staining in LP that contradicts results from Figure 1E.  This discrepancy should be clarified and inertness of LV-Vector justified. Thus, in addition, the clear effect on SA-β-gal staining from SCD2 moiety should be justified and not from LV-Vector as active moiety in LV-SCD2 construct.

 16.   Figure 5, 6 and 7 represents results of various “NC””and „mimic“ groups. The concepts and relevance of each NC and „mimic“ groups should be clearly described in respective 2.4, 2.5 and 2.6 sections or in the methodology section with corresponding references.  

 17.   Figure 5, 6 and 7 represent SA-β-gal staining in EP cells (in Figure 5) or LP cells (in Figure 6).  The type of cells in Figure 7 is not clearly defined. The reason for such variabnility is not explained.

 18.   The NC control is not provided in FIgures 6 and 7. This should be amend.

Reviewer 2 Report

Comments and Suggestions for Authors

In the present paper Yu et al addressed the role of SCD2 gene in MSC senescence. In particular, the authors studied the regulatory effect played by miR-200c-3p on SCD2 mRNAs and its contribution in replicative senescence.

Although the topic is of interest, some issues need to be addressed before considering the paper suitable for publication.

MAJOR ISSUES

1. The authors must refer to the MSC as Mesenchymal Stromal cells. This is the most appropriate name for defining the multipotent adult stem cell population with multi-lineage differentiative capacity (see PMID 37229499). Please modify through the text

2. Line 62: Please, explicit the acronym NSPCs; it is the first-time appearance within the text.

3. Line 79-82: the authors must add a brief hint about the role of miRNAs in chronic diseases, like cancer, and their potential as biomarkers (see PMID 35542114; 28671672).

4. The authors must properly edit the manuscript placing the “materials and methods” paragraph either following introduction or following discission.

5. The authors must provide more specifications about the constructs used for SCD2 and miR-200c-3p overexpression. They must rearrange the relative section in the “materials and methods” paragraph with all the information required.

6. Lines 135-138: The authors referred to previous data about the down-regulation of SCD2 protein expression levels in senescent cells. Although this evidence is widely acknowledged, they must provide the reference of their previous data. This is mandatory. Otherwise, the authors have to show the data they referred to.

7. It is widely acknowledged that there is not a unique marker assessing for cellular senescence. Rather than, the combination of specific markers is commonly used for discriminating senescent cells from normal ones. So, the authors must assay MSC senescence more in depth by corroborating the p16 expression levels with, at least, the p21 and p53 ones. Moreover, the proliferation status of the MSC must be assayed. Lastly, the ability of MSCs to differentiate into three lineages is a distinguishing characteristic for MSC senescence. Indeed, the imbalance in the differentiation into osteogenic and lipogenic lineages is a problem associated with MSC senescence along with the decrease in osteogenic differentiation and an increased tendency towards lipogenic differentiation (PMID: 38111850). Thus, the authors must test the adipocyte differentiation ability with appropriate staining (Oil Red) and suitable biomarkers (such as PPAR-γ and CEBPA). This approach will provide the correct evidence on MSc senescence in their model.

8. In my opinion, the authors must use of miR-200c-3p-inhibitor for rescue experiments. Only in this way they could corroborate the overexpression data.

Round 2

Reviewer 1 Report

Comments and Suggestions for Authors

Majority of responses and corrections made are reasonable and can be acceptable in principle.

Q8 and Q11 Responses. I agree with the reasoning for small sample size but can not beleave that such small sample size led always to normal distribution. Please provide data proving normality. Grafically with the results of normality test, preferable.

Reviewer 2 Report

Comments and Suggestions for Authors

In my opinion the authors did not addressed point 7 and point 8 properly.

The response to point 7 is weak. The authors should  test the senescence incidence in thier experimental setting. In the references provided they did not test the experimental hypothesis of the present paper. In my opinion, at least one more marker sholud be tested to corroborate their data

As concern point 8, I belive that tye lack of a proper rescue experiments must be included within the text as a limitation of the study.

Round 3

Reviewer 1 Report

Comments and Suggestions for Authors

THank you for your additional contributions.